# NL$^2$PS: A Natural Language to Lean Proofs System

**Yifan Luo**[*]
IIIS, Tsinghua University
2024311556
luoyf24@mails.tsinghua.edu.cn

**Kangping Xu**[*]
IIIS, Tsinghua University
2024311549
xkp24@mails.tsinghua.edu.cn

## 1   Background

The inference capabilities of large language models (LLMs) have garnered significant attention due to their potential for addressing complex scientific problems. With rapid advancements, these capabilities are now approaching the upper limits of current evaluation benchmarks. For instance, Llama3 [1] demonstrated remarkable improvements on MATH and GSM8k benchmarks, achieving score increases of 1690% and 668% over Llama1 [2] within just 1.5 years (from 2.9 to 51.9 and 11.0 to 84.5, 7&8B model respectively) [3, 4]. However, informal mathematical tasks, such as those in GSM8k (Grade School Math problems involving word-based puzzles), may not accurately reflect a model's reasoning abilities due to their high sensitivity to prompt formulation [5, 6].

To address these limitations, Lean—a proof assistant and functional programming language [7]—has emerged as a promising framework for evaluating and advancing LLM inference capabilities. Recent research [8, 9] integrating LLMs with Lean has generated significant interest in formal inference and automated theorem proving, demonstrating LLMs' potential to independently discover new theorems while maintaining robustness against prompt variations. Although current LLM performance on Lean-based benchmarks remains below that of informal benchmarks like GSM8k and MATH, there is an increasing need for a more comprehensive question set and a scalable, automated evaluation pipeline. These resources are crucial for rigorously assessing and advancing LLMs' reasoning capabilities within formal mathematical domains.

## 2   Definition

We describe the process of transforming an informal dataset into a formal one for use in Lean. Let $\mathcal{I}$ represent the informal dataset, consisting of problems and solutions in natural language. The formal dataset, $\mathcal{F}$, contains these problems translated into a formal language.

Each problem $x_i \in \mathcal{I}$ and its solution $y_i$ are mapped to formal representations using a function $F$. This formalization function $F : \mathcal{I} \rightarrow \mathcal{F}$ converts $(x_i, y_i)$ into a formal theorem and proof $(F(x_i), F(y_i))$.

Benchmark testing, denoted as $B$, evaluates the performance of language models by compiling their outputs. The compiler determines success with $C(\text{LLM}(F(x_i))) = 1$ if it compiles, and 0 otherwise. The accuracy is calculated as: Accuracy $= \frac{\sum_i^N C(\text{LLM}(F(x_i)))}{N}$, where $N$ is the total number of outputs. This method ensures an accurate assessment of the model's reasoning capabilities.

## 3   Related Work

**Benchmarks.**   To evaluate the theorem-proving capabilities of existing methods in Lean, several efforts have focused on gathering formal benchmarks from real competitions and textbook materials.

---

[*]Equal contribution.

Preprint. Under review.

The miniF2F dataset [10] collects high-school-level problems from AIME, AMC, and IMO, converting them into formal theorems and proofs. ProofNet [11] provides undergraduate-level informal theorems with their proofs but only includes the formal theorems without proofs. The FIMO benchmark [12], sourced from the IMO competition, is rarely used now due to its high difficulty and lack of support for the Lean 4 framework. PUTNAMBENCH [9] consists of formalized problems from the Putnam competition, targeting undergraduate students. Although there are a few benchmarks for evaluating Lean 4 theorem proving, they lack diversity in mathematical areas and difficulty levels, and the number of samples is insufficient. Additionally, these benchmarks struggle to elegantly handle certain problems in formalization, such as calculation problems, problems requiring the solver to independently find the answer, or those with commonsense elements in their natural language representation that are difficult to translate into formal language.

**Datasets.** In addition to benchmarks for evaluation, various datasets are utilized for training automated theorem provers. Some of these datasets are large-scale and synthesized through rejection sampling, using the Lean compiler on a set of seed problems. For example, AlphaProof [13] generated approximately 100 million formal problems using a method similar to AlphaZero. Deepseek-Prover [14] employs a Monte Carlo Tree Search (MCTS)-like strategy to create a training set of 9 million problems. Other datasets leverage existing Lean code authored by humans. LeanDojo [8] compiles 60,000 samples by collecting theorems from the Mathlib of Lean 4. LEAN-GitHub [15] aggregates Lean repositories from GitHub to extract usable theorems, resulting in a dataset of 28,000 samples.

## 4 Proposed Method

**Formalization.** Our formalization pipeline begins with human-annotated problems sourced from high school, competitions, and undergraduate textbooks. After filtering out unsuitable cases, such as theorems lacking proofs or problems heavily reliant on images, we will use LLMs like GPT-4o for formalization. To achieve high formalization accuracy, we plan to employ several strategies:

- **Few-shot prompting with RAG:** Inspired by ProofNet, we utilize Retrieval Augmented Generation (RAG) to retrieve the top-k informal problems with ground truth that closely resemble the target problem. This allows us to construct a few-shot prompt, enhancing accuracy through In-Context Learning (ICL).

- **Rejection sampling:** This strategy, commonly used in data selection, involves employing the Lean compiler to reject non-compiling code. The error information can then help the LLM modify the code for better results.

- **Backward translation:** Ensuring equivalence is crucial in auto formalization, as LLMs may simplify formal theorems by omitting important premises. To address this, we translate the formal representation back into informal problems and check for equivalence, preventing the loss of key elements. Moreover, with both the problem and reference answer, their formalization can be verified through a one-to-one correspondence.

**Research problems.** To generate equivalent formal representations in Lean, several research problems need to be considered:

- **Finding solutions:** Lean supports induction to work backward from the goal to the hypothesis, but this requires all theorems to have a well-defined goal. For arithmetic tasks or existence problems, the final answer must be explicitly presented as a goal. PUTNAMBENCH attempted to address this but only provided a workaround. We propose defining only the type of the goal, not the value, allowing the solution-finding process to occur within the Lean environment, which is highly verifiable without needing the goal to be predefined.

- **Progressive proving:** An important skill for a math expert is the ability to break down a complex proof into several lemmas, proving it step by step. We've noticed that some problems in our dataset have dependencies between subproblems or rely on other problems. Therefore, we plan to construct a subset to evaluate the prover's ability to utilize not only the standard library but also previously proven results in its context. This is a crucial skill for solving truly complex problems.

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
