# OpenReview forum: "N$\mathsf{L}^2$PS: A Natural Language to LEAN Proofs System"
_tsinghua.edu.cn/THU/2024/Fall/AML — THU 2024 Fall AML Submission_

### Official Review · ~Ethan_Wei_Yuxin1 · 2024-11-08
**Good job!**

**Rating:** 8
**Confidence:** 4

**Review:**

Your plan to address verification through equivalence checks and dependency management shows an impressive depth of foresight into the complexities of formal theorem proving.

The proposal demonstrates a strong foundation, and I look forward to seeing the results of your work on formalized datasets and Lean-based benchmarks. The choice of Lean and the structured pipeline you’ve outlined suggests your work could make a significant impact on the development of more robust, reasoning-capable language models.

---

### Official Review · ~王俊逸1 · 2024-11-08
**Advancing Formal Inference with NL2PS: A Bridge to Robust Mathematical Reasoning**

**Rating:** 8
**Confidence:** 4

**Review:**

The proposal "NL2PS: A Natural Language to Proofs System" presents a compelling vision for advancing the inference capabilities of large language models (LLMs) by bridging the gap between informal and formal mathematical reasoning. This initiative is particularly noteworthy for its potential to enhance the robustness of LLMs against prompt variations and to contribute to the discovery of new theorems in a formal, verifiable manner.

The authors' approach to transforming informal datasets into formal ones for use in Lean is both innovative and practical. By focusing on the creation of a scalable evaluation pipeline and a comprehensive question set, the proposal addresses a significant gap in current benchmarks, which often fail to accurately reflect a model's reasoning abilities due to their informal nature.

The proposed method's emphasis on formalization accuracy through few-shot prompting, rejection sampling, and backward translation is commendable. These strategies not only promise to improve the quality of formalized proofs but also ensure that the formal representations retain the essence of the original problems.

However, the proposal could benefit from a more detailed discussion on the challenges of formalization, especially when dealing with problems that require commonsense elements or have complex dependencies between subproblems. Additionally, while the focus on Lean as a proof assistant is well-founded, the integration of LLMs with such formal systems is still an emerging field, and the proposal could provide more insight into how it plans to navigate the technical complexities involved.

In summary, "NL2PS" is a forward-thinking proposal that has the potential to significantly advance the field of automated theorem proving and formal verification. It represents a strategic step towards developing LLMs that can reason with the precision and rigor of formal mathematical systems, offering a promising avenue for future research and practical applications in AI.

---

### Official Review · ~Zihan_Yan2 · 2024-11-10
**Very good!**

**Rating:** 9
**Confidence:** 4

**Review:**

The proposal introduces a system called NL2PS, designed to convert mathematical problems stated in natural language into formal proofs. This system leverages the reasoning capabilities of large language models (LLMs) and uses the Lean proof assistant and functional programming language to evaluate and enhance LLMs' capabilities in formal reasoning and automated theorem proving. This is a highly challenging problem. The proposal has a clear structure, a well-defined problem, and provides a detailed technical approach along with the challenges to be addressed.

---

### Official Review · ~Chan_Thong_Fong1 · 2024-11-10
**Compelling approach for mathematical problem-solving**

**Rating:** 9
**Confidence:** 4

**Review:**

The paper presents a compelling approach to bridging the gap between informal and formal mathematical problem-solving by leveraging the Lean proof assistant. It effectively highlights the limitations of current evaluation benchmarks like GSM8k and MATH, which are prone to prompt sensitivity, and argues for the robustness of formal verification through Lean-based methods. The pipeline is well-defined, incorporating strategies such as retrieval-augmented generation, rejection sampling, and backward translation to improve formalization accuracy. Overall, the paper makes a significant contribution by laying out a scalable, rigorous framework for evaluating LLMs' reasoning capabilities in formal mathematics.

---

### Official Review · ~Chentian_wei1 · 2024-11-10
**The paper offers a thorough account of NLP in math problem-solving with clear task definitions and detailed solutions, but it overly emphasizes datasets and benchmarks, and the evaluation criteria appear too simple.**

**Rating:** 9
**Confidence:** 4

**Review:**

The paper provides a detailed introduction to the research progress on natural language processing of mathematical problems, and the definition of the tasks is relatively clear. Detailed solutions are also presented point by point for the proposed methods. However, I believe the paper dedicates too much space to the introduction of datasets and evaluation benchmarks. Moreover, the current evaluation criteria seem somewhat simplistic.

---

### Official Review · ~Zhijie_shen3 · 2024-11-10
**Review of A Natural Language to LEAN Proofs System**

**Rating:** 10
**Confidence:** 4

**Review:**

### **Summary**
Overall, the paper presents a promising system that could push the boundaries of LLM capabilities in formal mathematical reasoning. Addressing the above suggestions would make the research proposal more comprehensive and impactful. If success this  proofs system could significantly advance the use of LLMs in formal mathematics, enabling better automation of theorem proving and potentially discovering new mathematical results

**Pros**
1.  The proposal introduces a proof system for converting natural language mathematical questions into formal Lean proofs. This is an innovative approach that addresses the limitations of current benchmarks, potentially leading to more accurate assessments of LLM capabilities.

2. The use of techniques like few-shot prompting, RAG, and rejection sampling demonstrates a thorough understanding of how to leverage existing LLMs to achieve the task. The combination of these techniques shows promise in improving the translation accuracy of natural language into formal proofs.

**Suggestions**
1. While the paper introduces the concept of a formalization function, it would benefit from a clearer explanation of how exactly the natural language inputs are transformed into Lean code. Providing specific examples of the conversion process could enhance understanding.

2. The current evaluation seems to rely solely on whether the Lean code compiles successfully.  And some is based on chatgpt4o capability, It would be helpful to include additional metrics, such as the complexity of the proofs generated or the model’s ability to generalize across different problem types.

---

### Official Review · ~Shuangyue_Geng1 · 2024-11-11
**Well-structured, but method clarity could be improved**

**Rating:** 9
**Confidence:** 4

**Review:**

This proposal presents a clear and impactful approach to developing NL2PS, a system for translating informal mathematical problems into formal proofs. The structured methodology, shows strong potential for advancing formalized reasoning in mathematics. However, the proposed methods would benefit from clearer, more accessible explanations to improve readability. Overall, this is a promising and well-organized project with minor refinements needed for clarity.

---

### Official Review · ~Yu_Zhang61 · 2024-11-11
**Review of "NPS: A Natural Language to LEAN Proofs System"**

**Rating:** 9
**Confidence:** 4

**Review:**

The NPS proposal introduces a system designed to convert natural language statements into Lean proofs, offering a novel means of assessing large language models (LLMs) within the realm of formal mathematics. By using Lean—a proof assistant valued for its rigor in formal logic—the project aims to go beyond current benchmarks, which often rely heavily on prompt formulation and informal reasoning tasks. The system integrates techniques like few-shot prompting, rejection sampling, and backward translation to ensure accurate formalization, with the ultimate goal of advancing automated theorem proving. This approach is ambitious and relevant, addressing critical gaps in current LLM evaluations that may overestimate reasoning abilities. However, the proposal would benefit from further elaboration on the specific challenges of formalization, such as handling ambiguities in informal language or the computational demands of real-time proof checking. Additionally, more information on evaluation criteria and expected outcomes would strengthen the proposal, enhancing its clarity and feasibility.

---

### Official Review · ~Jin_Zhu_Xu1 · 2024-11-11
**Clear Explanation and Comprehensive Related Work**

**Rating:** 8
**Confidence:** 4

**Review:**

The proposal points out a clear background motivation and expands a brief explanation about the roles of LLM as proof systems to improve benchmark models' reasoning abilities that are robust to prompt engineering. The proposal is well organized and well structured. However, it will be better by adding some clarification on the method explanation

---

### Official Review · ~Liu_Yiyang1 · 2024-11-11
**Clear, compelling and well elaborated proposal**

**Rating:** 9
**Confidence:** 4

**Review:**

This proposal outlines an approach to translate informal mathematical problems into formal proofs using Lean, a formal proof assistant. It introduces a novel approach by formalizing informal mathematical problems for use in Lean, an area where traditional LLM benchmarks fall short. The use of backward translation and rejection sampling adds robustness to the auto-formalization process, making this proposal particularly innovative in its methodological rigor. The proposal provides a well-defined formalization pipeline, and the inclusion of strategies like backward translation and rejection sampling demonstrates a thoughtful approach to achieving high formalization accuracy. Great job!

---

### Official Review · ~Changsong_Lei2 · 2024-11-12
**review of "NLPS: A Natural Language to LEAN Proofs System"**

**Rating:** 9
**Confidence:** 4

**Review:**

### Summary:
The proposal introduces NL2PS, a system to convert informal mathematical statements into formal proofs using the Lean proof assistant and large language models (LLMs), aiming to enhance the reasoning abilities of LLMs for formal mathematics by developing a robust formalization pipeline.

### Pros:
- Gives a clear defination to the problem and the proposed method. It sounds feasible.
- Uses techniques like retrieval-augmented generation, rejection sampling, and backward translation to improve the accuracy of converting informal problems into formal language.
- Proposes constructing diverse benchmark sets from various educational levels, which could significantly advance LLMs' capabilities in handling formal math reasoning tasks.

### Cons:
- Lacking detailed discussion about how the math problems are converted to Lean code.

---

### Official Review · ~Wuqian1 · 2024-11-12
**Review of "NL2PS: A Natural Language to Proofs System"**

**Rating:** 9
**Confidence:** 4

**Review:**

The proposal "NL2PS: A Natural Language to Proofs System" presents a robust approach to enhancing the inference capabilities of Large Language Models (LLMs) in the domain of formal mathematical reasoning. The quality of the work is evident in its comprehensive understanding of the limitations of current LLMs and the potential of leveraging Lean, a proof assistant, to overcome these limitations.
Pros
   1.Innovative Integration: The integration of LLMs with Lean for formal inference is a novel approach.
   2.Comprehensive Strategy: The proposed strategies for formalization are comprehensive and well-thought-out.
Cons
  1.Unproven in Practice: The effectiveness of the proposed strategies is yet to be proven in practice.
  2.Dependency on Lean: The success of the project is heavily dependent on the capabilities of the Lean proof assistant.

---

### Official Review · ~Yangchi_Gao1 · 2024-11-12

**Rating:** 9
**Confidence:** 4

**Review:**

The proposal presents an ambitious and potentially transformative approach to integrating natural language processing with formal mathematical reasoning. It has the potential to significantly advance the field of automated theorem proving and provide a more accurate assessment of LLM reasoning capabilities.

Advantages:
1.By focusing on formal mathematical domains, NL2PS addresses the limitations of current informal benchmarks, which may not accurately reflect a model's reasoning abilities due to their sensitivity to prompt formulation.
2.The use of few-shot prompting with RAG, rejection sampling, and backward translation to achieve high formalization accuracy is a well-thought-out strategy that leverages both human knowledge and LLM capabilities.

Disadvantages:
1.The formalization of natural language problems into formal proofs is a complex task, and the success of NL2PS may depend heavily on the quality of the LLMs used and the effectiveness of the formalization strategies.